# Radiosensitizer Effect of β-Apopicropodophyllin against Colorectal Cancer via Induction of Reactive Oxygen Species and Apoptosis

**DOI:** 10.3390/ijms222413514

**Published:** 2021-12-16

**Authors:** Jin-Hee Kwon, Na-Gyeong Lee, A-Ram Kang, Jie-Young Song, Sang-Gu Hwang, Hong-Duck Um, Joon Kim, Jong Kuk Park

**Affiliations:** 1Division of Radiation Biomedical Research, Korea Institute of Radiological and Medical Sciences, Seoul 01812, Korea; peterjane19canada@gmail.com (J.-H.K.); ilr27387@naver.com (N.-G.L.); arbam0919@kirams.re.kr (A.-R.K.); immu@kirams.re.kr (J.-Y.S.); sgh63@kirams.re.kr (S.-G.H.); hdum@kirams.re.kr (H.-D.U.); 2Division of Life Sciences, Korea University, Seoul 02841, Korea

**Keywords:** β-apopicropodophyllin, radiosensitizer, topoisomerase inhibitor, ROS, apoptosis, colorectal cancer

## Abstract

β-apopicropodophyllin (APP), a derivative of podophyllotoxin (PPT), has been identified as a potential anti-cancer drug. This study tested whether APP acts as an anti-cancer drug and can sensitize colorectal cancer (CRC) cells to radiation treatment. APP exerted an anti-cancer effect against the CRC cell lines HCT116, DLD-1, SW480, and COLO320DM, with IC50 values of 7.88 nM, 8.22 nM, 9.84 nM, and 7.757 nM, respectively, for the induction of DNA damage. Clonogenic and cell counting assays indicated that the combined treatment of APP and γ-ionizing radiation (IR) showed greater retardation of cell growth than either treatment alone, suggesting that APP sensitized CRC cells to IR. Annexin V–propidium iodide (PI) assays and immunoblot analysis showed that the combined treatment of APP and IR increased apoptosis in CRC cells compared with either APP or IR alone. Results obtained from the xenograft experiments also indicated that the combination of APP and IR enhanced apoptosis in the in vivo animal model. Apoptosis induction by the combined treatment of APP and IR resulted from reactive oxygen species (ROS). Inhibition of ROS by N-acetylcysteine (NAC) restored cell viability and decreased the induction of apoptosis by APP and IR in CRC cells. Taken together, these results indicate that a combined treatment of APP and IR might promote apoptosis by inducing ROS in CRC cells.

## 1. Introduction

Colorectal cancer (CRC) is the third most common type of cancer in the United States, with almost 150,000 patients newly diagnosed with CRC in 2020. Although usually occurring in individuals aged over 50 years, about 12% of newly diagnosed patients and 7% of deaths from CRC have been reported in individuals aged less than 50 years [1,2]. The treatment modalities for CRC include surgery, radiation therapy, and systemic treatments such as chemotherapy, immunotherapy, and targeted therapy including the anti-EGFR (epidermal growth factor receptor) agent cetuximab and the anti-angiogenesis agent bevacizumab. Most patients with CRC are diagnosed with regional or distant metastases, requiring additional chemotherapy after surgery or palliative chemotherapy for metastatic cancer [3,4]. Traditional chemotherapy agents and radiotherapy for CRC affect not only the tumor but also non-target organs, resulting in the development of adverse effects or resistance [5]. Resistance is a major obstacle to cancer treatment, being responsible, directly or indirectly, for over 80% of deaths; therefore, novel drugs and treatment strategies are required to overcome tumor resistance to current treatments.

Podophyllotoxin acetate (PA) is a natural compound that induces cancer cell death by disturbing microtubule stability and inducing cell cycle arrest, ER stress, and autophagy. This makes this molecule a candidate for novel cancer drug development [6]. PA is a chemical agent related to podophyllotoxin (PPT), which was isolated from *Podophyllum peltatum Linnaeus* and shown to have antiviral affects against herpes, measles, and influenza viruses, as well as against venereal warts and skin cancer. PPT has been involved in the development of semi-synthetic anti-cancer derivatives, such as etoposide, teniposide and etopophos, which act by inhibiting DNA topoisomerases [7]. DNA topoisomerases are one of the major chemotherapeutic targets, and continuous efforts have been made to develop novel antibacterial and anticancer agents [8,9]. The topological modifications related to DNA transcription and replication are controlled by various topoisomerases through several mechanisms such as by introducing negative or positive supercoils into the DNA, catenating or decatenating circular and linear DNA, and relaxing positive or negatively supercoiled DNA [10,11]. There are three categories of molecular mechanisms for topoisomerase inhibition. One of them involves the competitive inhibition of the binding of substrates directly to the active site of topoisomerases. The other mechanisms include competitive inhibition at the ATP binding site and blocking the enzyme activity by inducing the formation of a ternary protein–DNA–inhibitor complex [12,13,14,15,16]. Among these inhibitory mechanisms, topoisomerase-targeted chemotherapy is postulated as clinically significant because through this mechanism, the enzyme is forced to form a cleavage complex, causing cell death [17]. Another topoisomerase-targeted chemotherapy includes a redox-productive effect and the induction of covalent bond formation in the drug–enzyme complex, which is followed by cell death. [18]. These topoisomerase inhibitor-derived anti-cancer drugs, such as irinotecan and etoposide, are used for chemotherapy of several cancers including non-small cell lung cancer (NSCLC) and colorectal cancer (CRC). [19,20,21]. β-apopicropodophyllin (APP) is a derivative of PPT that disturbs microtubule polymerization and induces apoptosis of NSCLC cells such as NCI-H460, NCI-H1299, A549, and their xenografts in nude mice in in vivo models, suggesting that it may be a candidate anti-cancer agent, as reported in our previous studies [22,23]. To further prove the anti-cancer effect of APP, thus extending our previous findings, the anti-cancer or radiosensitizing effect of APP against CRC cells in vitro and in vivo were evaluated in the present study.

## 2. Results

### 2.1. APP Induces the Death of CRC Cells by Inducing DNA Damage

Previously, APP (Figure 1A) was synthesized from PPT and shown to have anti-cancer and radiosensitizing effects against NSCLC cells in vitro and in vivo [17,18]. This study tested whether APP could also induce cell death and have radiosensitizing effects against two CRC cell lines, HCT116 and DLD-1. The IC50 values of APP after 48 and 72 h were calculated to be 9.79 nM and 8.51 nM, respectively, for HCT116 cells and 9.27 nM, and 8.41 nM for DLD-1 cells (Figure 1B–E). We also detected the IC50 values of APP for additional CRC cell lines, i.e., SW480 and COLO320DM. The IC50 values of APP after 48 and 72 h were 13.78 nM and 9.84 nM, respectively, for SW480 and 8.28 nM and 7.76 nM, respectively, for COLO320DM. These IC50 values for various CRC cells were used as standard doses in the following experiments, based on the assumption that APP might be an effective anti-cancer drug for treating colorectal cancer. Because HCT-116 cells contain wild type *p53* and DLD-1 cells contain mutant *p53* [24,25], the similar IC50 values in these cell lines indicated that the cytotoxicity of APP is not related to *p53* status. To confirm the anti-cancer effect of APP against CRC cells, immunoblot analysis was performed using an antibody recognizing γH2AX. APP treatment was found to increase the expression of γH2AX in CRC cells (Figure 1C), suggesting that the anti-cancer effect of APP is due to its induction of DNA damage.

### 2.2. APP Acts as a Radiosensitizer by Retarding Cell Growth In Vitro

The radiosensitization effects of APP in HCT116 and DLD-1 cells were tested in clonogenic assays. We used HCT116 and DLD-1 to test the *p53*-independent radiosensitizing effect of APP, because the HCT116 cell line contains the wild-type *p53* gene, but DLD-1, SW480 and COLO320DM contain the mutant *p53* gene [26]. Cells were pre-treated with 7.5 nM APP for 16 h, followed by IR with 1, 2, 3, or 4 Gy and culture of the cells for 14 days. Clonogenic assays showed that the surviving fraction of cells treated with APP and IR was lower than the surviving fraction of cells treated with IR alone (Figure 2A). Based on a survival fraction of 0.1, the dose enhancement ratios (DERs) for HCT-116 and DLD-1 cells were calculated to be 1.13 and 1.31, respectively (Table 1). A DER value >1 indicated that APP enhanced the effects of radiation. Additionally, cell counting assays were performed to assess the effects of the combined treatment of 7.5 nM APP and 3 Gy IR on cell death (Figure 2B–E). Cell death rates were over 1.5–2-fold higher in HCT-116, DLD-1, SW480, and COLO320DM cells treated with APP and IR than in cells treated with APP or IR alone (Figure 2B–E). In addition, immunoblot analyses with the anti-γH2AX antibody showed that the combined treatment of APP and IR enhanced γH2AX expression in all cell lines compared with treatment with APP or IR alone (Figure 2F). Collectively, these results indicate that APP can induce radiosensitization in CRC cells possibly by inducing DNA damage, which leads to an anti-cancer effect.

### 2.3. Combined Treatment with APP and IR Increases Apoptosis

To identify the cell death pathway induced by APP radiosensitization, we first assessed whether the combined treatment with APP and IR could induce apoptosis in HCT-116 and DLD-1 cells (Figure 3A). Using an Annexin V–PI apoptosis detection kit I, we found that apoptotic cell death in both CRC cell lines was more than two-fold greater following the combined treatment with APP and IR than after treatment with APP or IR alone. Immunoblot analyses also showed that the levels of cleaved caspase-3, caspase-9, and PARP were higher in cells treated with the combination of APP and IR compared with those in cells treated with APP or IR alone (Figure 3B). Immunoblot analysis of the mitochondrial and cytosol fractions of these cells also showed that the concentration of cytochrome *c* in the cytosol was higher following treatment with APP and IR than after either treatment alone (Figure 3C), indicating that the combined treatment with APP and IR enhanced cell death by activating the apoptosis pathway.

### 2.4. The Radiosensitizing Effect of APP Is Mediated by Intracellular ROS Accumulation

Because the disruption of ROS homeostasis has been shown to be a main cause of radiosensitizer-induced apoptosis in our previous studies [27,28], we also performed H2DCFDA-based intracellular ROS detection assays. These assays showed that the production of ROS was 1.3–1.5 fold higher in cells treated with APP and IR than in cells treated with APP or IR alone (Figure 4A). Treatment with the ROS scavenger NAC abrogated the increase in ROS production induced by the combined treatment with APP and IR (Figure 4B). To test whether the NAC-associated reduction in ROS reduced cell death following the combined treatment with APP and IR, cell counting and Annexin V–PI apoptosis assays were performed, both of which showed that NAC pre-treatment prior to the combined treatment with APP and IR inhibited the induction of cell death (Figure 5A,B). Immunoblot analyses also showed that NAC pre-treatment reduced γH2AX expression, the release of cytochrome *c* into the cytosol, and the cleavage of PARP and caspases induced by the combined treatment with APP and IR (Figure 6A–C). These findings indicate that the DNA damage and cell death induced by the combined treatment with APP and IR were due to the accumulation of ROS in both CRC cell lines.

### 2.5. In Vivo Radiosensitization Effect of APP

Finally, we tested whether APP radiosensitized tumors by inducing apoptosis in vivo. HCT116 cells were injected into BALB/c nude mice, and xenograft tumors were treated intra-tumorally with 5 mg/kg APP, 5 Gy IR, both, or neither. Each xenograft tumor was harvested 30 days later, and TUNEL assays were performed to detect apoptotic cells (Figure 7A). Measurements of the percentages of cells that had undergone apoptosis in each group (Figure 7A, right panel) and the relative ratios of stained (apoptotic) areas (Figure 7A, left panel) showed that the percentages of dead cells in the control, APP-treated, IR-treated, and APP + IR-treated tissue samples were 4.9%, 26.1%, 50.2%, and 74.2%, respectively, with statistical analyses showing that the combined treatment with APP and IR enhanced apoptosis compared with the treatment with APP or IR alone. In addition to showing that APP could enhance apoptosis in vivo, these findings showed that APP could induce apoptosis in vitro and in vivo by enhancing DNA damage and ROS production in CRC cells (Figure 7B).

## 3. Discussion

In this study, we demonstrated that APP induced the expression of phospho-H2AX (γH2AX), a major biomarker for the DNA damage response (DDR) activated by several DNA damage inducers such as IR that can recruit DNA repair proteins [29,30,31]. H2AX is a component of histone proteins in chromatin that is rapidly phosphorylated at the serine-139 position by ataxia telangiectasia mutated (ATM) and ATM-Rad3-related (ATR) kinases, yielding γH2AX, in response to various DNA damaging reagents [32,33]. The induction of γH2AX by APP suggests that APP might damage DNA in CRC cells. We also found that APP induced radio-sensitization of CRC cells. Both clonogenic and cell counting assays showed that APP enhanced IR-induced cell growth retardation, with the combined treatment with APP and IR also increasing γH2AX induction. These results showed that APP may be a candidate radiosensitizer as well as a candidate anti-cancer drug against CRC cells. The combined treatment with APP and IR also enhanced apoptosis in HCT116 and DLD-1 CRC cells by inducing the release of mitochondrial cytochrome *c* into the cytosol. We also observed that in vivo treatment of tumor xenografts with the combined treatment of APP and IR enhanced apoptotic cell death more than three-fold. These results indicated that the combined treatment of APP and IR enhanced the apoptosis induction. Two major pathways of apoptosis have been identified: the extrinsic and the intrinsic pathways [34,35]. The extrinsic pathway begins with death receptor/ligand binding [36,37] and proceeds through the activation of the initiator caspases-8 and -10. Activated cleaved caspases sequentially induce the cleavage of the effector caspases-3 and -7, inducing apoptosis [33]. The intrinsic pathway is initially induced by intracellular stresses, such as DNA damage, ER stress, hypoxia, and metabolic stress. These stresses lead to changes in mitochondrial outer membrane permeabilization and the release of cytochrome *c*, which interacts with apoptotic protease activating factor 1 (APAF1), thereby inducing apoptosome assembly and the activation of caspase-9 [38,39]. Active caspase-9 continuously activates caspases-3 and -7 to induce apoptosis. Caspases-8 and -9 act as initiator caspases in the extrinsic and intrinsic pathways, respectively, resulting in the activation of caspase-3, a common ‘executioner’ caspase in both apoptotic pathways. The BH3-only protein BH3-interacting death domain agonist (BID) provides crosstalk between the extrinsic and the intrinsic apoptotic pathways by inducing caspase-8 cleavage [40]. Our immunoblotting data results demonstrate the activation of caspases-3 and -9 and the release of mitochondrial cytochrome *c*, results consistent with the involvement of the intrinsic apoptosis pathway. Because the activation or modulation of apoptosis-related proteins, such as p53, p21, Bcl-2, Bax and caspases, constitutes one of the strategies for radiotherapy-induced cancer elimination, therapeutic reagents that enhance the effects of radiotherapy should also target the apoptosis machinery in cells [41,42,43]. Moreover, we found that the combined treatment of APP and IR increased ROS production, whereas treatment with the ROS scavenger NAC decreased ROS induction. Enhanced ROS was associated with enhanced apoptotic cell death, accompanied by γH2AX induction and cytochrome *c* release. These three effects of APP and IR—apoptosis, DNA damage, and mitochondrial outer membrane disruption—were likely due to ROS production, as these effects were reversed by treatment with NAC. ROS may be an especially important mediator of the radiosensitization effects of various regents, including APP, in several types of cancer [22,27,28]. Radiosensitizers identified to date include small molecules, such as free radicals and pseudosubstrates, nanomaterials, including gold-based nanometallic materials with flexible surface-engineered molecular structures and favorable kinetic properties that promote radiosensitization, and macromolecules, such as miRNAs, peptides, proteins, and oligonucleotides. Several small-molecule chemicals that enhance free radical production, such as APP, have been developed as radiosensitizers [44]. The therapeutic effect of conventional radiation treatment is usually mediated by the indirect action of free radicals produced by the radiolysis of water, followed by the destruction of biomolecules. These effects can be enhanced by small molecules that promote free radical production [45]. Taken together, these findings indicate that APP is a promising radiosensitizer candidate that induces apoptosis via DNA injury and the production of free radicals such as ROS.

## 4. Materials and Methods

### 4.1. Cell Cultures and Chemical Reagents

HCT116, DLD-1, SW480, and COLO320DM cell lines were purchased from the American Type Culture Collection (Rockville, MD, USA). The cell lines were cultured in RPMI 1640 medium (Corning, Manassas, VA, USA) supplemented with 10% fetal bovine serum (FBS; Gibco, Grand Island, NY, USA) and 1% streptomycin/penicillin (Wellgene, Gyeongsan-si, South Korea) at 37 °C in a 5% CO_2_ incubator. β-apopicropodophyllin (5-(3,4,5-trimethoxyphenyl)furo(3′,4′:6,7)naphtho(2,3-d)(1,3)dioxol-6(9H)-one)) was synthesized by J&C Sciences (Daejeon, Korea).

### 4.2. MTT Assay and IC50 Determination

HCT116, DLD-1, SW480, and COLO320DM cells were seeded on 96-well plates (3 × 10^3^ cells/well) and incubated with various concentrations (6.25, 12.5, 25, 50, 100 nM) of APP for 48 or 72 h at 37 °C. A 20 μL aliquot of MTT (3-(4,5-dimethylthiazol-2-yl)-2,5-diphenyltetrazolium bromide) solution (2 mg/mL) was added to each well, and the cells were incubated for 1 h at 37 °C. Formazan crystals generated in living cells were dissolved in 100 μL of DMSO, and the absorbance of individual wells at 545 nm was measured using a microplate reader (Molecular Devices, San Jose, CA, USA). The 50% inhibitory concentration (IC50) of APP was calculated by a concentration–response analysis using Softmax Pro software Ver. 6.5 (Molecular Devices, Sunnyvale, CA, USA).

### 4.3. Clonogenic Assay

HCT116 and DLD-1 cells were seeded in 60 mm dishes at concentrations that yielded 20–100 colonies/dish (100, 200, 400, 600, 1000 cells/dish). After incubation for 24 h, the cells were treated with or without 7.5 nM APP for 16 h and irradiated with 1, 2, 3, or 4 Gy ^137^Cs as a source of γ-ionizing radiation (Atomic Energy of Canada, Ltd., Mississauga, ON, Canada). The cells were incubated for a maximum of 14 days until colonies formed, and colonies larger than 200 μm in diameter were stained with 1% methylene blue in methanol. Stained colonies were counted using a colony counter (Imaging Products, Chantilly, VA, USA). The number of colonies per dish was calculated relative to the number of cells seeded per dish, and the dose enhancement ratio (DER) of each cell line was determined from the numbers of colonies using Excel software (Microsoft Co., Redmond, WA, USA).

### 4.4. Cell Counting Assay

HCT116, DLD-1 cells, SW480, and COLO320DM (1 × 10^5^ cells/60 mm dish) were pre-treated or not with 7.5 nM APP for 16 h and then exposed to 3 Gy irradiation. The cells were incubated for 72 h at 37 °C, collected by trypsinization, washed twice with cold PBS, and stained with trypan blue. The cells were counted using an EVE^TM^ Automated Cell Counter (NanoEntek, Seoul, Korea). The results are reported as the mean of triplicate assays.

### 4.5. Immunoblot Analysis

The harvested cells were lysed with RIPA buffer (50 mM Tris-HCl, pH 7.6, 150 mM NaCl, 1% Triton X-100, 1% sodium deoxycholate, 0.1% SDS, 2 mM EDTA), and the lysates were centrifuged at 14,000× *g* for 20 min. The supernatants were removed, and the concentration of proteins measured at 280 nm, using the Bradford solution (Bio-Rad, Hercules, CA, USA) and a microplate reader (Molecular Devices). A 20 mg aliquot of lysate from each well was loaded onto an SDS-PAGE gel, electrophoresed, and transferred to a nitrocellulose membrane. After blocking for 1 h with 5% skim milk, the membranes were incubated at 4 °C with primary antibodies in 5% BSA solution at 4 °C, followed by washing and incubation with secondary antibodies for 1 h at room temperature. Bands were detected with Clarity^TM^ Western ECL Substrate (Bio-Rad) and visualized with Amersham ImageQuant 800 (GE Healthcare Bio-Sciences Corp., Marlborough, MA, USA). The primary antibodies used in this study included antibodies to pro-caspase-3, pro-caspase-9, cleaved caspase-3, cleaved caspase-9, pro-PARP, and cleaved PARP (Cell Signaling Technology, Beverly, MA, USA). An anti-β-actin antibody (Sigma-Aldrich, St. Louis, MO, USA) was used as a loading control.

### 4.6. Isolation of Mitochondrial and Cytosolic Fractions

HCT116 and DLD-1 cells were pre-treated with 7.5 nM APP and irradiated at 3 Gy, followed by incubation for 48 h at 37 °C. The cells were incubated with trypsin–EDTA for 5 min, collected by centrifugation at 250 g for 1 min, resuspended in extraction buffer (1 M sucrose, 1 M HEPES, pH 7.4, 1 M KCl, 1 M MgCl2, 0.25 M EGTA, and 1 M DTT), homogenized, and centrifuged at 12,000× *g* at 4 °C for 15 min. The supernatants were regarded as the cytosolic fractions. Each pellet was resuspended in lysis buffer (50 mM Tris-HCl, pH 7.6, 150 mM NaCl, 1% Triton X-100, 1% sodium deoxycholate, 0.1% SDS, 2 mM EDTA) and regarded as the mitochondrial fraction. The fractions were immunoblotted with an anti-cytochrome *c* antibody, with VDAC used as a loading control. These experiments were performed in triplicate.

### 4.7. Annexin V–Propidium Iodide Assay

HCT116 and DLD-1 cells (1 × 10^5^ cells/60 mm dish) were treated with 7.5 nM APP or left untreated and exposed to 3 Gy IR. The cells were incubated for 72 h, collected by trypsinization, washed twice with cold PBS, and stained with FITC Annexin V apoptosis detection kit I reagent (Becton Dickinson, Franklin Lakes, NJ, USA), as described in the manufacturer’s protocol. The samples were loaded onto a FACSort flow cytometer (Becton Dickinson), and the fraction of apoptotic cells was measured (*x*-axis, FL1 channel; *y*-axis, FL-2 channel). These experiments were repeated in triplicate.

### 4.8. ROS Detection

HCT116 and DLD-1 cells (5 × 10^5^ cells/60 mm dish) were pre-treated with 7.5 nM APP or left untreated, exposed to 3 Gy IR, and incubated for 24 h. The cells were subsequently stained with 25 mM H2DCFDA (Merck, Darmstadt, Germany) for 5 min, trypsinized, harvested by centrifugation, and resuspended in cold PBS. The samples were loaded onto a FACSort flow cytometer (Becton Dickinson) for detection and analysis of intracellular ROS (*x*-axis, FL1 channel; *y*-axis, Counts).

### 4.9. TUNEL Assay of Xenografts

The protocols of all animal experiments were approved by the Institutional Animal Care and Use Committee (IACUC No. kirams 2019-0064). Xenografts were generated by subcutaneously injecting 1 × 10^7^ HCT116 cells/mouse into 6-week-old BALB/cAnNCrj-nu/nu mice (Envigo, Cambridgeshire, UK). The mice were divided into four groups of four mice each, with one group each treated with APP alone, IR alone, APP + IR, and no treatment (control). When the xenografts reached 100−120 mm^3^ in size, 5 mg/kg APP in DMSO was intratumorally injected into the mice in the APP-only and APP + IR groups, with an equal volume of DMSO (vehicle) intratumorally injected into mice in the IR-only and control groups. Six hours later, mice in the IR-only and APP + IR groups were locally irradiated with 5 Gy IR using a ^60^Co γ-ray source (Theratron 780; AECL Ltd., Mississauga, ON, Canada). The treatments were repeated three times at 3-day intervals for a total of 12 days. The mice were sacrificed 30 days after the start of the experiment. Extracted tumors were fixed with formaldehyde, embedded in paraffin, and sectioned. TUNEL assays for detection of dUTP nicks were performed by Super Bio Chips Co. (Seoul, Korea). Each stained tissue sample was photographed with an Olympus BX 53 (Olympus, Shinjuku, Tokyo, Japan), and the ratios of TUNEL-positive to total cells in each image were measured with Image J software (NIH, Bethesda, MD, USA). The percentages of TUNEL-positive cells in each tissue sample were calculated relative to the number of cells in the control group.

### 4.10. Statistical Analysis

Results are reported as means ± standard deviations (SDs), shown as error bars, and compared using Student’s *t*-tests. All statistical analyses were performed using GraphPad Prism software Ver. 5 (GraphPad Software, La Jolla, CA, USA), with *p*-values < 0.05 considered statistically significant.

## Figures and Tables

**Figure 1 ijms-22-13514-f001:**
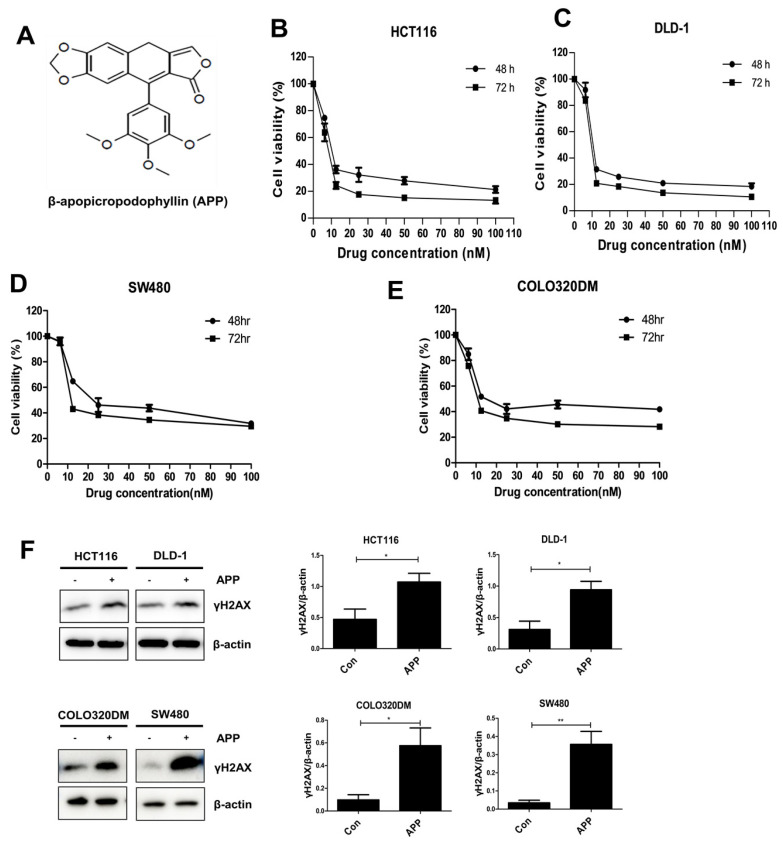
Anti-cancer effects of APP in human colorectal cancer cells. (**A**) Chemical structure of APP. (**B**–**E**) IC50 values determination for APP. The CRC cell lines HCT116, DLD-1, SW480, and COLO320DM cells were treated with 6.25, 12.5, 25, 50, or 100 nM APP for 48 or 72 h, and cell viability was measured by the MTT assay. IC50 values of APP for CRC cell lines were calculated as described in the Materials and Methods. (**F**) Immunoblot assays for the detection of γH2AX activation in cells treated with APP. HCT116, DLD-1, SW480, and COLO320DM cells were incubated with or without 7.5 nM APP for 24 h prior to harvesting. The relative band densities were determined via densitometry using ImageJ software (NIH, Bethesda, MD, USA) and then normalized to that of each control. All experiments were repeated in triplicate, and statistical analyses were performed with Student’s *t*-test; * *p* < 0.05, ** *p* < 0.01. Bands in the figures show representative data.

**Figure 2 ijms-22-13514-f002:**
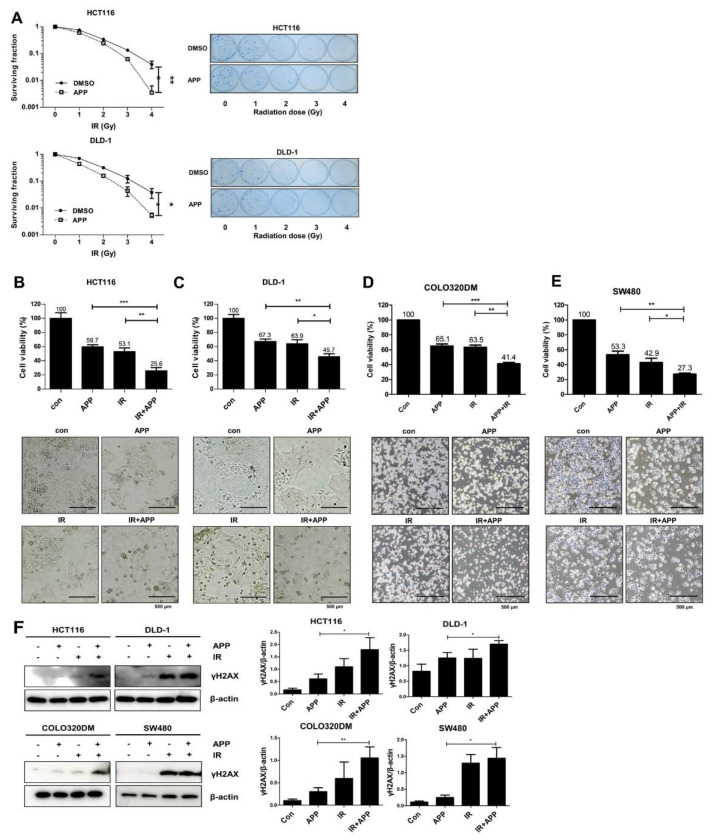
Radiosensitizer effect of APP. (**A**) Clonogenic assays. Clonogenic assays for DLD-1 and HCT116 cells were performed as described in the Materials and Methods. ‘DMSO’, DMSO-treated mock control; ‘APP’, cells treated with 7.5 nM APP. (**B**–**E**) Cell counting assay. Cell counting assays were performed as described in the Materials and Methods. ‘Con’, DMSO-treated mock control cells; ‘APP 7.5 nM’, cells treated with 7.5 nM APP; ‘IR 3Gy’, cells treated with 3Gy IR; ‘IR + APP’, cells treated with 7.5 nM APP and 3Gy IR. The lower panel shows microscopic images prior to cell detachment. Experiments were repeated in triplicate, and the results indicate the mean of triplicate assays [22,27]. Each bar in the pictures indicates 500 μm. (**F**) Immunoblot assay for γH2AX. HCT116, DLD-1, SW480, and COLO320DM cells were incubated with or without 7.5 nM APP, exposed to 3 Gy IR, and incubated for 72 h prior to harvest. Each graph in the right panel indicates the statistical analysis of immunoblot bands. The relative band densities were determined via densitometry using ImageJ software (NIH, Bethesda, USA) and then normalized to the density of each control. All experiments were repeated in triplicate, and statistical analyses were performed with Student’s *t*-test; * *p* < 0.05, ** *p* < 0.01, *** *p* < 0.001. Bands in the figures show representative data.

**Figure 3 ijms-22-13514-f003:**
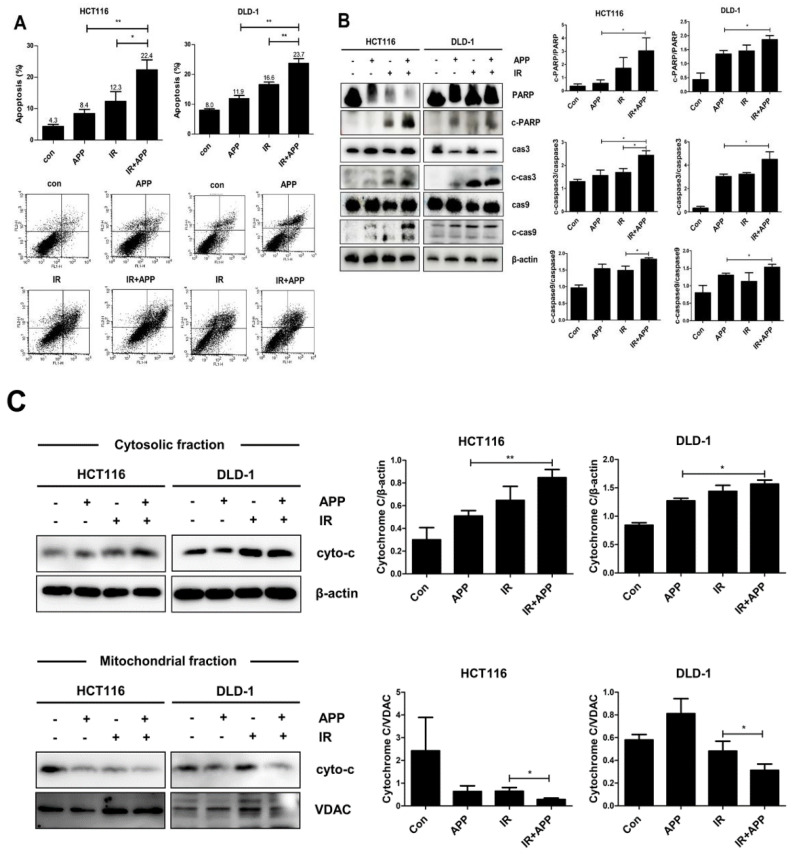
Induction of apoptotic cell death by the combined treatment with APP and IR in CRC cells. (**A**) Annexin V–PI assay for the detection of apoptosis. HCT116 and DLD-1 cells were treated with 7.5 nM APP alone, 3 Gy IR alone, or both for 72 h. The assay method is described in the Materials and Methods. The lower panel shows representative FACSort flow cytometry images prior to quantitative analyses. ‘Con’, DMSO-treated mock control cells; ‘APP 7.5 nM’, cells treated with 7.5 nM APP; ‘IR 3Gy’, cells treated with 3Gy IR; ‘IR + APP’, cells treated with 7.5 nM APP and 3Gy IR. Experiments were repeated in triplicate, and the results indicate the mean of triplicate assays. (**B**) Immunoblot assays. The assays detected caspase-3, caspase-9, and PARP activation in cells treated with APP and IR; ‘cas3′, caspase-3; ‘cas9′, caspase-9; ‘c-cas3′, ‘c-cas9′ and ‘c-PARP’ indicate the cleaved forms of caspases-3 and -9 and PARP. HCT116 and DLD-1 cells (1 × 10^5^) were incubated with or without 7.5 nM APP, exposed to 3 Gy IR, and incubated for 72 h prior to harvest. Each graph in the right panel indicates the statistical analysis of the immunoblot bands. (**C**) Immunoblot assays. Immunoblot assay for detecting the release of cytochrome *c* from mitochondria to cytosol in cells treated with APP and IR. HCT116 and DLD-1 cells (3 × 10^5^) were incubated with or without 7.5 nM APP for 16 h and then exposed to 3 Gy IR for 72 h. Each graph in the right panel indicates the statistical analysis of the immunoblot bands. The relative band densities were determined via densitometry using ImageJ software (NIH, Bethesda, USA) and then normalized to that of each control. All experiments were repeated in triplicate, and statistical analyses were performed with Student’s *t*-test; * *p* < 0.05, ** *p* < 0.01. Bands in the figures show representative data.

**Figure 4 ijms-22-13514-f004:**
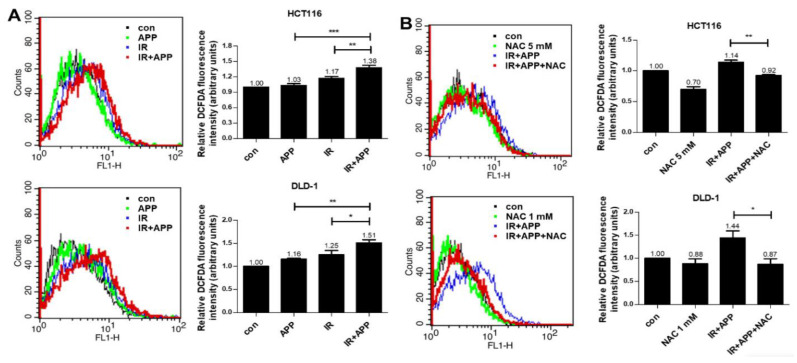
Enhancement of ROS accumulation in CRC cells treated with APP and IR. (**A**) Detection of ROS production. ROS detection in cells treated with APP alone, IR alone, or both by FACSort flow cytometry. HCT116 and DLD-1 cells (5 × 10^5^) were treated with or without 7.5 nM APP for 16 h and then exposed to 3 Gy IR for 24 h prior to harvesting. ‘Con’, DMSO-treated mock control cells; ‘APP’, cells treated with 7.5 nM APP; ‘IR’, cells treated with 3Gy IR; ‘IR + APP’, cells treated with both 7.5 nM APP and 3Gy IR. Experiments were repeated in triplicate, and the results indicate the mean of triplicate assays. (**B**) Effects of NAC on ROS production. HCT116 cells were pre-treated with 5 mM NAC for 2 h, and DLD-1 cells were pre-treated with 1 mM NAC for 1 h, before treatment with APP, IR, or both. ‘Con’, DMSO-treated mock control cells; ‘APP’, cells treated with 7.5 nM APP; ‘IR’, cells treated with 3Gy IR; ‘IR + APP’, cells treated with 7.5 nM APP and 3Gy IR; ‘IR + APP + NAC’, cells pre-treated with NAC and treated with 7.5 nM APP and 3Gy IR. Experiments were repeated in triplicate, and the results indicate the mean of triplicate assays. Statistical analyses were performed with Student’s *t*-test; * *p* < 0.05, ** *p* < 0.01, *** *p* < 0.001.

**Figure 5 ijms-22-13514-f005:**
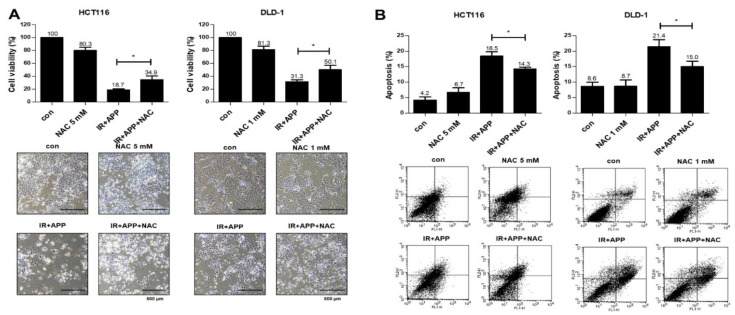
Role of ROS in the radiosensitizing effect of APP. (**A**) Cell counting assay. HCT116 or DLD-1 cells (5 × 10^5^) were treated with or without 7.5 nM APP and then exposed to 3 Gy IR for 24 h prior to harvest. In addition, HCT116 cells were pre-treated with 5 mM NAC for 2 h, and DLD-1 cells were treated with 1 mM NAC for 1 h, before the combined treatment with APP and IR. ‘Con’, mock control cells; ‘NAC’, cells pre-treated with 5 or 1 mM NAC; ‘IR + APP’; cells treated with 7.5 nM APP and 3Gy IR; ‘IR + APP + NAC’, cells pre-treated with NAC and treated with 7.5 nM APP and 3Gy IR. Each lower panel shows representative microscopic images prior to cell detachment. Bars in cell pictures indicate 500 μm. Experiments were repeated in triplicate, and the results indicate the mean of triplicate assays. Statistical analyses were performed with Student’s *t*-test; * *p* < 0.05. (**B**) Annexin V–PI assay for the detection of apoptosis. Samples were prepared as described in (**A**), and apoptosis rates for each condition were determined by flow cytometry. Experiments were repeated in triplicate, and the results indicate the mean of triplicate assays. Each lower panel indicates representative FACSort flow cytometry images prior to quantitative analyses. Scale bar: 500 μm.

**Figure 6 ijms-22-13514-f006:**
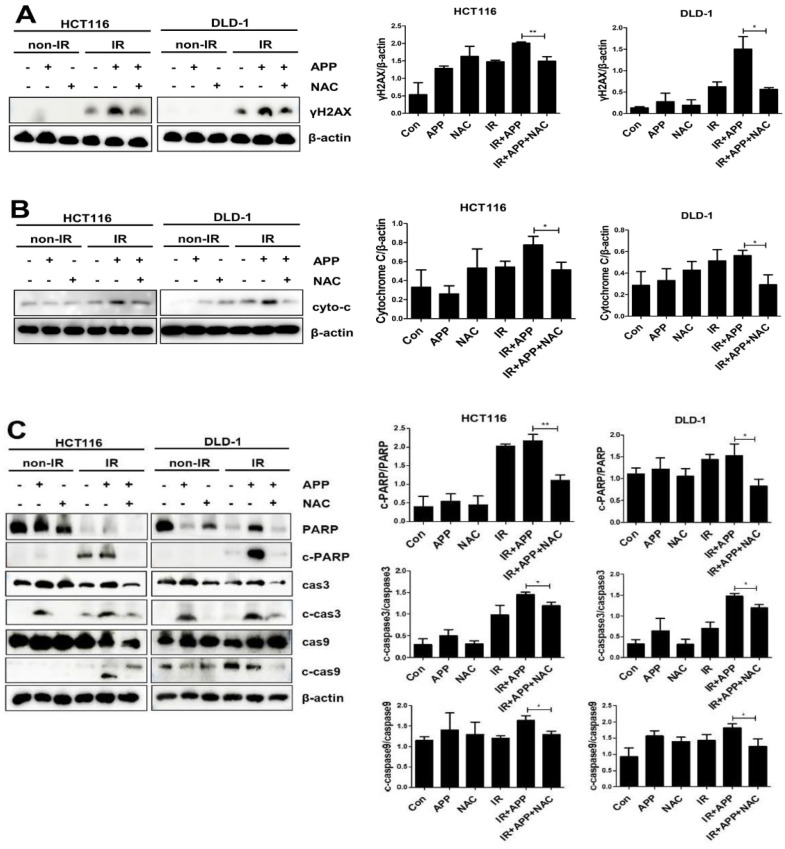
Effect of NAC on the activation of cell death-related proteins. HCT116 or DLD-1 cells (5 × 10^5^) were pre-treated with 5 mM NAC for 2 h or with 1 mM for 1 h, respectively, followed by treatment with or without 7.5 nM APP, 3 Gy IR, or both. Immunoblot assays were performed for the detection of (**A**) γH2AX activation, (**B**) cytochrome *c* release, and (**C**) apoptosis proteins. Activated caspase-3, caspase-9, and PARP were detected in cells pre-treated with NAC and treated with APP and IR for 72 h. ‘cas3′, caspase-3; ‘cas9′, caspase-9; ‘c-cas3′, cleaved caspase-3; ‘c-cas9′, cleaved caspase-9; ‘c-PARP’, cleaved PARP. The relative band densities in each right panel were determined via densitometry using ImageJ software (NIH, Bethesda, USA) and then normalized to that of each control. All experiments were repeated in triplicate, and statistical analyses were performed with Student’s *t*-test; * *p* < 0.05, ** *p* < 0.01. Bands in the figures show representative data.

**Figure 7 ijms-22-13514-f007:**
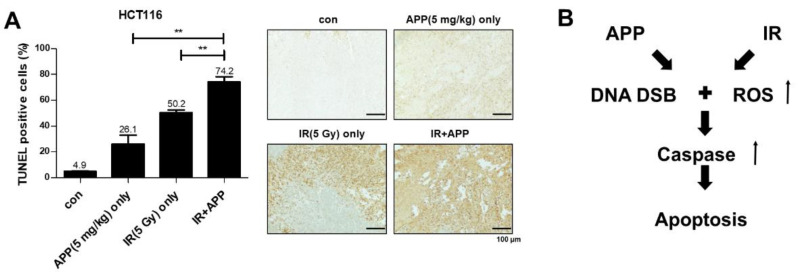
In vivo radiosensitizing effects of APP. (**A**) TUNEL assays measuring apoptotic cells in mouse xenografts. Mice injected with HCT116 cells were divided into four groups, a control group, an APP-only group (5 mg/kg), an IR-only group (5 Gy), and an APP+ IR group, in which mice were pretreated with 5 mg/kg APP for 6 h prior to IR irradiation. Each group included five mice, and apoptotic cells in each xenograft were detected by the TUNEL assay. The dark brown area in each tissue sample, indicating staining with the TUNEL reagent, was detected using Image J software Ver.1.8.0. The graph indicates quantitative analyses of the ratios of TUNEL-stained apoptotic cells to total cells. ‘Con’, mock control; ‘APP only’, mice treated with 5 mg/kg APP; ‘IR only’, mice treated with 5 Gy IR; ‘IR + APP’, mice treated with 5 mg/kg APP and 5 Gy IR. Each bar in the figures of tissues indicates 100 μm. Experiments were repeated in triplicate, and the results indicate the mean of triplicate assays. Statistical analyses were performed with Student’s *t*-test; ** *p* < 0.01. (**B**) Schematic diagram showing the radiosensitization effect of APP. Scale bar: 100 μm.

**Table 1 ijms-22-13514-t001:** DER analysis.

	DER (Dose Enhancement Ratio)
	7.5 nM
HCT116	1.13
DLD-1	1.31

Note: DER (Dose Enhancement Ratio) values were calculated from the clonogenic assay (Figure 2A). Radiation doses corresponding to a survival fraction of 0.1 for the mock control or for the 7.5 nM APP-treated group were acquired for HCT116 cells and resulted to be 3.16 and 2.79 Gy, respectively; for DLD-1 cells, they were 3.28 and 2.51 Gy, respectively. Using these doses, DER values were calculated as follows: radiation doses at a survival fraction of 0.1 of the IR-only treatment group/radiation doses at a survival fraction of 0.1 of the APP-treated group.

## Data Availability

The data presented in this study are available on request from the corresponding author.

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
