# Peer review of "Radiosensitizer Effect of β-Apopicropodophyllin against Colorectal Cancer via Induction of Reactive Oxygen Species and Apoptosis"

_ijms, 2021, doi:10.3390/ijms222413514_

Round 1

Reviewer 1 Report

Figure 1F:  Immunoblot assays for detection of γH2AX activation in cells treated with APP was only performed on DLD-1 and HCT-116. Similar experiments should be done on COLO320DM and SW480 lines. Same issue with Figure 2F, Table 1, Figure 3, Figure 4.

Author Response

We agreed Reviwer’s comments and performed additional immunoblot assays for detection of γH2AX activation using COLO320DM and SW480 cell lines which was added as Figure 1F and 2F. These data show that radiosensitizing effect of APP resulted from increased expressions of γH2A that leads to enhanced DNA damage.

Data in Table 1, Figure 3 and Figure 4 indicate radiosensitizing mechanism of APP.

In these experiments, we used 2 cell lines (HCT116 and DLD-1)for mechanism studies, because HCT116 contains wild type p53 gene while DLD-1, SW480, COLO320DM express mutant p53 gene (Ahmed, D.; Eide, P.W., Eilertsen, I.A.; Danielsen, S.A.; Eknæs, M.; Hektoen, M.; Lind, G.E.; Lothe, R.A. Epigenetic and genetic features of 24 colon cancer cell lines.  Oncogenesis. 2013, 2, e71). We added descriptions on these data into 2.2 and added new reference 26

Reviewer 2 Report

The paper by Lee and co-workers entitled “Radiosensitizer effect of β-apopicropodophyllin against colorectal cancer via induction of reactive oxygen species and apoptosis” investigates the anti-cancer potential and radio-sensitizing effects of a podophyllotoxin derivative in in vitro models of colorectal cancers, as well as in a xenograft model in mice, where β-apopicropodophyllin proved to enhance the sensitivity of cancer cells, both in vitro and in vivo, to gamma-radiation as well as, increasing cell death and apoptosis, via a the induction of oxidative stress.

The research is well performed and follows a logic thread. Nevertheless, there are some points that need to be addressed before the paper can be considered for publication.

1) In line 38, since podophyllotoxin is known from centuries, considering it as a “novel candidate” may sound discordant. Therefore, Authors should change this sentence.

2) Overall, the Introduction is quite confusing. Therefore, I suggest Authors to critically read this section to improve its readability. For example, the passage regarding the modes of inhibition of topoisomerase are difficult to comprehend. Moreover, the aim of the study is usually at the end of this section, rather than in the middle (lines 67-69). Please, improve this part.

3) Why did Authors choose to perform some assays in HCT116, DLD-1, SW480 and COLO320DM cell lines, while all the other tests were performed just on the first two?  Please discuss the reasons underlining this choice in the paper to help the reader understand.

4) Please, soften the sentence at line 100. The evidence is too scarce to claim such thing.

5) Please, show the meaning of each statistic sign in each figure as well the number of replicates to understand the strength of the results.

6) Table 1 is not mentioned in the text. Please, solve this issue and include the other two cell lines since, as said in point 3, it is not clear why from this point on of the paper Authors chose to work with HCT116 and DLD-1 cells.

7) Why did Authors choose not to test the combination of NAC and APP without gamma radiation to prove the pro-oxidant mechanism of action of APP. In my opinion, it would have been a more appropriate control to compare results to.

8) In the Discussion, there is no mention to the in vivo results. Please, solve this issue.

Author Response

The paper by Lee and co-workers entitled “Radiosensitizer effect of β-apopicropodophyllin against colorectal cancer via induction of reactive oxygen species and apoptosis” investigates the anti-cancer potential and radio-sensitizing effects of a podophyllotoxin derivative in in vitro models of colorectal cancers, as well as in a xenograft model in mice, where β-apopicropodophyllin proved to enhance the sensitivity of cancer cells, both in vitro and in vivo, to gamma-radiation as well as, increasing cell death and apoptosis, via a the induction of oxidative stress.

The research is well performed and follows a logic thread. Nevertheless, there are some points that need to be addressed before the paper can be considered for publication.

1) In line 38, since podophyllotoxin is known from centuries, considering it as a “novel candidate” may sound discordant. Therefore, Authors should change this sentence.

-> We agree with your comment, and changed the sentence as “making it as a candidate for novel cancer drug development”

2) Overall, the Introduction is quite confusing. Therefore, I suggest Authors to critically read this section to improve its readability. For example, the passage regarding the modes of inhibition of topoisomerase are difficult to comprehend. Moreover, the aim of the study is usually at the end of this section, rather than in the middle (lines 67-69). Please, improve this part.

-> i) We re-organized Introduction according to your comment.

  1. ii) References were re-numbered according to the re-organized introduction. The last sentence of Introduction (line 84-87) was deleted: “Our results show that radio-sensitization and anti-cancer effect of APP is due to apoptosis caused by DNA damage and reactive oxygen species (ROS) production”

3) Why did Authors choose to perform some assays in HCT116, DLD-1, SW480 and COLO320DM cell lines, while all the other tests were performed just on the first two?  Please discuss the reasons underlining this choice in the paper to help the reader understand.

 -> In these experiments, we used 2 cell lines - HCT116 and DLD-1 - for mechanism studies, because HCT116 contains wild type p53 gene, while DLD-1, SW480, and COLO320DM show mutant p53 gene expression (Ahmed, D.; Eide, P.W., Eilertsen, I.A.; Danielsen, S.A.; Eknæs, M.; Hektoen, M.; Lind, G.E.; Lothe, R.A. Epigenetic and genetic features of 24 colon cancer cell lines.  Oncogenesis. 2013, 2, e71). We added new paragraph into 2.2 and added new reference 26.

4) Please, soften the sentence at line 100. The evidence is too scarce to claim such thing.

 -> We exchanged the sentence:

“APP is effective anti-cancer drug treating colorectal cancer.”-> “APP might be an effective anti-cancer drug treating colorectal cancer.”

5) Please, show the meaning of each statistic sign in each figure as well the number of replicates to understand the strength of the results.

-> We added the sentence: “Experiments are repeated in triplicate and results indicate the mean of triplicate assays” in legends of Figure 2B-E, Figure 3A, Figure 4A and 4B, Figure 5A and 5B, Figure 7.

6) Table 1 is not mentioned in the text.

- > We added following sentence into 2.2: “More than 1 of DER values indicated APP enhanced radiation effects [22, 28].”

Please, solve this issue and include the other two cell lines since, as said in point 3, it is not clear why from t his point on of the paper Authors chose to work with HCT116 and DLD-1 cells.

-> In these experiments, we used 2 cell lines, HCT116 and DLD-1, for mechanism studies, because HCT116 contains wild type p53 gene, while DLD-1, SW480, and COLO320DM show mutant p53 gene expression (Ahmed, D.; Eide, P.W., Eilertsen, I.A.; Danielsen, S.A.; Eknæs, M.; Hektoen, M.; Lind, G.E.; Lothe, R.A. Epigenetic and genetic features of 24 colon cancer cell lines.  Oncogenesis. 2013, 2, e71). We added a description on this into 2.2 and added new reference 26.

7) Why did Authors choose not to test the combination of NAC and APP without gamma radiation to prove the pro-oxidant mechanism of action of APP. In my opinion, it would have been a more appropriate control to compare results to.

-> We focused on radiosensitizing effect of APP in this study, and performed immunoblotting for observation of cell stress and cell death signaling in Figure 6. As shown in Figure 6, we validated combination of NAC and APP without gamma radiation did not exert cell stress and cell death in our system.

8) In the Discussion, there is no mention to the in vivo results. Please, solve this issue.

-> We agree and added the sentence into Discussion:

Xenograft assay data performing TUNEL staining also indicated enhancement of apoptotic cell death by combination of IR and APP (Fig. 7A).

Reviewer 3 Report

The manuscript of Jong Kuk Park and co-authors is an interesting study on β-apopicropodophyllin antitumour properties; it follows two other papers from the same research group and it is one of the few recent studies on this topic. Since lately it is a high interest towards podophyllotoxin derivates application as antitumour prodrugs, I suggest introducing this term as well among the keywords of the manuscript, to ensure a better visibility.

It is a well-written paper, which connect the in vitro data with the animal studies, in order to elucidate the redox mechanisms behind the β-apopicropodophyllin radiosensitizer effects.  Reproducible experimental details were provided, and the results are clearly presented, including the good quality figures and explicit figure captions.

The cytotoxicity was measured on four cell lines; authors should explain why they chose DLD-1 and HCT116 cell lines for further testing. In chapter 2.4 it is missing the combined effect between NAC+APP and NAC+IR; please insert in the figures or explain why these are not relevant.

In the discussion of the manuscript, I suggest to compare the results with similar in vitro or in vivo publications on standard neoadjuvant drugs used together with irradiation in colorectal cancer (5-FU, oxaliplatin, or bevacizumab). As well, etoposide vs. β-apopicropodophyllin parallels could be of interest, the drug being a semisynthetic  Podophyllum peltatum derivate.   The inhibition of DNA-topoisomerases is mentioned several times in the introduction, but no discussion was found about possible associations between the obtained results and topoisomerase inhibition. If they were correlations between the APP in vitro and in vivo radiosenzitizing effect, some details should be given.   

Author Response

The manuscript of Jong Kuk Park and co-authors is an interesting study on β-apopicropodophyllin antitumour properties; it follows two other papers from the same research group and it is one of the few recent studies on this topic. Since lately it is a high interest towards podophyllotoxin derivates application as antitumour prodrugs, I suggest introducing this term as well among the keywords of the manuscript, to ensure a better visibility.

It is a well-written paper, which connect the in vitro data with the animal studies, in order to elucidate the redox mechanisms behind the β-apopicropodophyllin radiosensitizer effects.  Reproducible experimental details were provided, and the results are clearly presented, including the good quality figures and explicit figure captions.

The cytotoxicity was measured on four cell lines; authors should explain why they chose DLD-1 and HCT116 cell lines for further testing. In chapter 2.4 it is missing the combined effect between NAC+APP and NAC+IR; please insert in the figures or explain why these are not relevant.

  • We agree your suggestion. We focused radiosensitizing effect of APP in this study, and performed immunoblotting for observation of cell stress and cell death signaling in Figure 6. As shown in Figure 6, we validated combination of NAC and APP without gamma radiation did not exerted cell stress and cell death in our system.

In the discussion of the manuscript, I suggest to compare the results with similar in vitro or in vivo publications on standard neoadjuvant drugs used together with irradiation in colorectal cancer (5-FU, oxaliplatin, or bevacizumab). As well, etoposide vs. β-apopicropodophyllin parallels could be of interest, the drug being a semisynthetic  Podophyllum peltatum derivate.   The inhibition of DNA-topoisomerases is mentioned several times in the introduction, but no discussion was found about possible associations between the obtained results and topoisomerase inhibition. If they were correlations between the APP in vitro and in vivo radiosenzitizing effect, some details should be given.   

  • We agree your suggestion. We already examined human topoisomerase IIα activity using Human Topoisomerase II assay kit (come from TopoGEN Co.) in group treated with APP, but, unfortunately, we could not acquire clear results from inhibition of topoisomerase activity.

Round 2

Reviewer 1 Report

I have noticed some improvements in the manuscript in response to previous review cycle. However, it is hard to believe that the authors have conducted the experiments (in particular western blots) multiple times (considering the short period of time between the previous review and the resubmission). The figure legends do not mention how many times the western blots were conducted. Densitometric analysis of ALL western blots (performed on multiple independent experiments) is required to have a more meaningful analysis of data. 

Author Response

I have noticed some improvements in the manuscript in response to previous review cycle. However, it is hard to believe that the authors have conducted the experiments (in particular western blots) multiple times (considering the short period of time between the previous review and the resubmission). The figure legends do not mention how many times the western blots were conducted. Densitometric analysis of ALL western blots (performed on multiple independent experiments) is required to have a more meaningful analysis of data. 

-> Our immunoblot assays were repeated in twice. We detected densities of all bands and calculated mean values, and then acquired to relative ban densities that marked on each bands.

We also added the sentences to 4.5.

: The relative band densities were determined via densitometry using ImageJ software (NIH, Bethesda, USA) and then normalized to that of each control. All experiments were repeated twice and calculated to mean values. Bands in the figures signify representative data.

Reviewer 2 Report

Authors have thoroughly addressed each point. My only comment is to remove the mention of Figure 7A in the added sentence in the Discussion section, since it is not advisable to refer to figures in this part of the manuscript.

Author Response

Authors have thoroughly addressed each point. My only comment is to remove the mention of Figure 7A in the added sentence in the Discussion section, since it is not advisable to refer to figures in this part of the manuscript.

-> We removed the sentence form Discussion

: Xenograft assay data performing TUNEL staining also indicated enhancement of apoptotic cell death by combination of IR and APP (Fig. 7A).

Round 3

Reviewer 1 Report

In previous versions the authors had not mentioned how many times the experiments were conducted. Now, two days after the last review, the authors claim that the westerns were performed twice.  

Author Response

In previous versions the authors had not mentioned how many times the experiments were conducted. Now, two days after the last review, the authors claim that the westerns were performed twice.  

-> We performed immunoblot analyses in triple and statistical analyses of each experiment.

-> Results of statistical analyses were indicated with graphs in Fig. 1F, 2F, 3B, 3C, 6A-C. Related sentences were inserted into each figure legends.

-> Raw images of western bands in triple were added as a Supplementary data.

-> Additionally, we also inserted sentence about statistical analyses of animal tissues in Fig. 7 legend.

-> We also exchanged author order of Na-Gyeong Lee and Jin-Hee Kwon.

Round 4

Reviewer 1 Report

No further comments/suggestions by this reviewer. 

This manuscript is a resubmission of an earlier submission. The following is a list of the peer review reports and author responses from that submission.

Round 1

Reviewer 1 Report

The manuscript is interesting and the study design seems appropriate.

However, some issues need to be addressed before publication.

The authors should carefully revise the language, style and grammar of the manuscript because several sentence and grammar errors were found.

line 53: is "topoisomerase poisons phenomenon" considered a scientific term?;

lines 57-60: the authors should mention the names of the "topoisomeraase inhibitor-derived anti-cancer drugs" used against NSCLC and CRC;

line 62: the authors should mention what type of tumor model was used in their previous studies. Was it cell lines, animal models? Please specify;

lines 70-71: the authors mention that one of the systemic treatments used to treat CRC is hormonal therapy. This information is wrong and the authors should correct this. Moreover, the authors fail to mention the targeted therapies that are also used in CRC treatment;

lines 298-304: the authors need to include the company names and references of the cell culture media and supplements used in the cell culture.

Author Response

The manuscript is interesting and the study design seems appropriate.

However, some issues need to be addressed before publication.

The authors should carefully revise the language, style and grammar of the manuscript because several sentence and grammar errors were found.

line 53: is "topoisomerase poisons phenomenon" considered a scientific term?;

->We agreed your comments. We exchanged “topoisomerase poisons phenomenon” into “topoisomerase-targeted chemotherapy”.

lines 57-60: the authors should mention the names of the "topoisomeraase inhibitor-derived anti-cancer drugs" used against NSCLC and CRC;

-> We agreed your comments. We modified “topoisomerase inhibitor-derived anti-cancer drugs” into “topoisomerase inhibitor-derived anti-cancer drugs such as irinotecan and etoposide. ”.

line 62: the authors should mention what type of tumor model was used in their previous studies. Was it cell lines, animal models? Please specify;

-> We agreed your comments. We added “induces apoptosis of NSCLC cells” into “induces apoptosis of NSCLC cells such as NCI-H460, NCI-H1299, A549, and their xenografts in nude mice in vivo”.

lines 70-71: the authors mention that one of the systemic treatments used to treat CRC is hormonal therapy. This information is wrong and the authors should correct this. Moreover, the authors fail to mention the targeted therapies that are also used in CRC treatment;

->We agreed your comments. We deleted “hormonal therapy” from our manuscript, and added “targeted therapy such as anti-EGFR (epidermal growth factor receptor) agent cetuximab and the anti-angiogenesis agent bevacizumab)”. Therefore, we also inserted new Reference 22.

lines 298-304: the authors need to include the company names and references of the cell culture media and supplements used in the cell culture.

-> We agreed your comments, and then added company names and references of the cell culture media and supplements.

Reviewer 2 Report

The research manuscript titled Radiosensitizer effect of β-apopicropodophyllin against colorectal cancer via induction of reactive oxygen species and apoptosis by Lee et al., is interesting. The authors have used adequate number of relevant reference articles to support their claims and have used a series of coherent experiments.  It would increase the interest of the scientific community if the authors could use additional cell lines (or patient samples) to show the broad applicability of the data presented. 

Author Response

The research manuscript titled Radiosensitizer effect of β-apopicropodophyllin against colorectal cancer via induction of reactive oxygen species and apoptosis by Lee et al., is interesting. The authors have used adequate number of relevant reference articles to support their claims and have used a series of coherent experiments.  It would increase the interest of the scientific community if the authors could use additional cell lines (or patient samples) to show the broad applicability of the data presented. 

-> Thank you for your helpful comments. Unfortunately, the editorial office allowed us to finish the revision in 5 days, which is not enough time to investigate on different cell lines or human clinical samples. We will try to extend our study according to your comments in our next issues.

Round 2

Reviewer 2 Report

The research article titled “Radiosensitizer effect of β-apopicropodophyllin against colorectal cancer via induction of reactive oxygen species and apoptosis” by lee et al., examines whether the podophyllotoxin derivative, β-apopicropodophyllin (APP), can lower the apoptosis threshold of colorectal cancer cells to apoptotic effects of radiation therapy. The authors have used two colorectal cancer cell lines (CRC) and have shown the IC50 values of APP against these two lines.

They have further shown that APP in combination with γ-ionizing radiation, compared to each agent alone, reduces the growth of CRC lines. They are further claiming that the increased apoptosis is due to increased ROS production as ROS inhibition by NAC could restore cell viability. Finally, the authors conclude that combination treatment CRC lines with APP and IR may promote apoptosis by increased generation of ROS.

The results for PARP cleavage in Figure 3C and 6 are not convincing. No PARP cleavage by combination treatment in HCT-11 line and the basal levels of PARP in DLD-1 line is significantly higher in combination treatment compared to untreated cells. The non-cleaved and cleaved forms of PARP and caspases should be shown on the same blot.

Figure 6A the western blot results for HCT-11 are not convincing. The levels are b actin are not equal. And the levels of H2Ax for both lines are unclear and the experiments must be repeated.

The results of animal studies are interesting.  The authors have used appropriate references to support their idea. The other major issue is that the authors have only used two cell lines. In order to show the general applicability of the combination treatment proposed by the authors, additional cell lines must be used.

Author Response

The research article titled “Radiosensitizer effect of β-apopicropodophyllin against colorectal cancer via induction of reactive oxygen species and apoptosis” by lee et al., examines whether the podophyllotoxin derivative, β-apopicropodophyllin (APP), can lower the apoptosis threshold of colorectal cancer cells to apoptotic effects of radiation therapy. The authors have used two colorectal cancer cell lines (CRC) and have shown the IC50 values of APP against these two lines.

They have further shown that APP in combination with γ-ionizing radiation, compared to each agent alone, reduces the growth of CRC lines. They are further claiming that the increased apoptosis is due to increased ROS production as ROS inhibition by NAC could restore cell viability. Finally, the authors conclude that combination treatment CRC lines with APP and IR may promote apoptosis by increased generation of ROS.

The results for PARP cleavage in Figure 3C and 6 are not convincing. No PARP cleavage by combination treatment in HCT-11 line and the basal levels of PARP in DLD-1 line is significantly higher in combination treatment compared to untreated cells. The non-cleaved and cleaved forms of PARP and caspases should be shown on the same blot.

  • We agreed your comments, and suggest apoptosis induction by APP radiosensitizing effect are demonstrated and compensated with caspases detections in Figure 3B and 6C. Therefore, we deleted PARP immnoblot data (Part of Figure 3B and 6C) because its cleavages are not convincing. And we also deleted its related sentences and original images in manuscript and supplementary data.

Figure 6A the western blot results for HCT-11 are not convincing. The levels are b actin are not equal. And the levels of H2Ax for both lines are unclear and the experiments must be repeated.

  • We agreed your comments, and suggest DNA damage inductions by APP radiosensitizing effect are compensated with Figure 2C, and ROS effects on apoptosis with Figure 5 and 6. We deleted not convincing data - H2AX and β-actin immnoblot data (Figure 6A) and its related sentences and original images in manuscript and supplementary data.

The results of animal studies are interesting.  The authors have used appropriate references to support their idea. The other major issue is that the authors have only used two cell lines. In order to show the general applicability of the combination treatment proposed by the authors, additional cell lines must be used.

  • We agreed your comments, but we already confirm radiosensitizer effect of APP in various genetic backgrounds –containing non-small cell lung cancer cells (Ref. 17. H460 is wild type p53 and PTEN, but H1299 cell is deleted p53 and PTEN). And we used two-kinds of genetic backgrounds CRC cells – HCT116 and DLD-1. Especially, HCT116 contains mutant p53 but DLD-1 is wild type p53 (Oncogenesis. 2013 Sep; 2(9): e71.). Therefore, we suggest radiosensitizing effect of APP already demonstrated without additional cell line experiments.

Round 3

Reviewer 2 Report

The authors have not addressed any of comments made by this reviewer. Rather than repeating the experiments showing basal levels and cleaved forms of caspases 3 and 9 as well as PARP, the authors have simply removed the data about PARP and have not provided western blots for caspases showing basal levels and cleaved forms of caspases on the same blot. These antibodies are available and the experiments are rather simple to perform. Additionally, the authors have not provided any new blot where the actin levels are identical.

Lastly, the authors have not provided data using additional cell lines as requested.